# Does Body Mass Index Impact the Clinical Response to Dupilumab Therapy in Atopic Dermatitis? A Monocentric Study of 170 Patients

**DOI:** 10.3390/jcm13154559

**Published:** 2024-08-05

**Authors:** Selene Nicolosi, Francesca Barei, Maurizio Romagnuolo, Silvio Fumagalli, Angelo Valerio Marzano, Silvia Mariel Ferrucci

**Affiliations:** 1Unit of Allergology and Immunology, ASST Grande Ospedale Metropolitano Niguarda, 20162 Milan, Italy; 2Dermatology Unit, Fondazione IRCCS Ca’ Granda Ospedale Maggiore Policlinico, 20122 Milan, Italy; francesca.barei@policlinico.mi.it (F.B.);; 3Department of Medical Biotechnologies, University of Siena, 53100 Siena, Italy; 4Independent Researcher, 23900 Lecco, Italy; 5Department of Pathophysiology and Transplantation, Università Degli Studi di Milano, 20122 Milan, Italy

**Keywords:** atopic dermatitis, biological therapy, body mass index (BMI), dupilumab, weight

## Abstract

**Background**: Dupilumab is a monoclonal antibody used for the treatment of moderate/severe atopic dermatitis (AD). In recent years, several studies have confirmed the positive association between AD and overweight/obesity, and a report demonstrated the effect of weight reduction on the improvement of AD symptoms. **Methods**: The weight of 170 patients under treatment with dupilumab was recorded at baseline and after 48 weeks (T48). Clinical monitoring was mainly conducted using the Eczema Area and Severity Index (EASI). The study aimed to assess a possible correlation between the clinical outcome of dupilumab therapy and BMI. **Results**: Although not statistically significant, patients with a BMI < 25 have a higher EASI percentage improvement than patients with a BMI ≥ 25 at any time point, and the percentage of overweight and obese patients that does not reach EASI-75 at T48 is higher compared to normal-weight patients (13.5% vs. 5.9%). Despite this, in the multivariate regression analysis, no baseline characteristic, including BMI, appears to increase the risk of not reaching EASI-75. In addition, the results show no differences in BMI between baseline and T48 in any age/sex group. **Conclusions**: The results suggest that overweight and obese patients have a lower response to dupilumab when considering the EASI score, but this difference does not appear to be clinically significant. Furthermore, dupilumab treatment does not seem to impact weight.

## 1. Introduction

Atopic dermatitis (AD) is one of the most common chronic illnesses worldwide, affecting 3.5% of the global population (230 million people). Although it can manifest at any age [1], its incidence is highest in infancy, with around 80% of cases appearing before the age of 6 [2,3]. Clinically, AD is characterized by eczematous lesions that vary with age, yet it is also highly heterogeneous in terms of severity, progression, and sometimes specific clinical features [4]. A meta-analysis of seven birth cohort studies with follow-ups of up to 26 years suggests that the annual prevalence of AD in adults in developed countries can reach as high as 14.6%, confirming that AD is a lifelong condition [5]. A systematic review of 378 studies published globally from 1958 to 2018 found that the overall prevalence of AD in children is between 0.96% and 22.6%, while the overall prevalence of AD in adults is between 1.2% and 17.1% [6].

Interleukin (IL)-13 and IL-4 are crucial contributors to the inflammation in AD, resulting in chronic type-2 inflammation. Dupilumab is a humanized IgG4 monoclonal antibody that targets the IL-4 receptor alpha chain (IL-4Rα), which is present in both type 1 (IL-4 specific) and type 2 (IL-4 and IL-13 specific) IL-4 receptor complexes, thus inhibiting both IL-4 and IL-13 signaling. Dupilumab was the first biological therapy approved for the treatment of atopic dermatitis. Its efficacy has been proven in both clinical trials and real-life studies [7,8,9,10,11,12,13,14,15,16].

To the best of our knowledge, only a single Swedish study described a possible association between weight gain and dupilumab: in a cohort of 12 AD patients treated with dupilumab, all patients experienced significant weight gain after 48 weeks of follow-up [17]. The authors speculated that since IL-4α receptor signaling is essential for the development of postnatal brown fat, inhibiting IL-4 and IL-13 might interfere with its activation, potentially raising the risk of obesity in AD patients.

Overweight and obesity are defined by a body mass index (BMI) of 25 to 29.9 kg/m^2^ and a BMI of ≥30 kg/m^2^, respectively. In recent years, several studies have confirmed a positive association between AD and overweight/obesity in both infancy and adulthood [18,19]. Additionally, a report showed that weight reduction positively impacts the improvement of AD symptoms. [20]. The link between inflammatory dermatosis and obesity can be attributed to the fact that adipose tissue in obese individuals induces a systemic inflammatory state, resulting in altered serum levels of cytokines, chemokines, and adipokines. In particular, the level of leptin, an adipokine with numerous proinflammatory effects, is increased in AD patients [21], while adiponectin, which seems to have an anti-inflammatory role, is found at lower levels in patients with AD [21,22]. Furthermore, obesity has been demonstrated to compromise the epidermal barrier, causing increased transepidermal water loss and skin dryness [23,24].

## 2. Materials and Methods

### 2.1. Population

The study population consists of patients with AD under treatment with dupilumab, followed clinically by the Allergological Dermatology Service at the Dermatology Unit of the Fondazione IRCCS Ca’ Granda Ospedale Maggiore Policlinico of Milan (Italy). This is a single-center retrospective study. Ethical approval is referred to as protocol Dupi Long Term 2022. All the patients have given written informed consent for the publication of their case details. We collected the data from 170 Caucasian adult patients with severe AD who were treated with dupilumab at standard doses (600 mg at baseline, then 300 mg every other week) for 48 weeks. Patients who did not reach the 48-week follow-up were excluded.

### 2.2. Data Collection

We collected data on sex, age, height, baseline weight, atopic comorbidities, AD phenotype, atopic family history, AD onset age, the previous use of systemic drugs for AD, and the intake of systemic drugs for AD at baseline. Additionally, the patient’s weight was recorded after 48 weeks of treatment. AD phenotypes were classified according to Salvador et al. [25]. The onset age was further classified as follows: infants (0 years old), children (1–12 years old), adolescents (13–17 years old), and adults (≥18 years old). Patients were divided into two age groups at baseline: 18–49 years old and ≥50 years old. The population was also divided into two groups based on baseline BMI: patients with a BMI < 25 (normal weight) and patients with a BMI ≥ 25 (overweight and obese). For clinical scores, we considered the Eczema Area and Severity Index (EASI) and the following patient-reported outcomes (PROs): Pruritus Numerical Rating Scale (NRS), sleep NRS, Patient-Oriented Eczema Measure (POEM), Dermatology Life Quality Index (DLQI), and Atopic Dermatitis and Control Tool (ADCT). Data were collected at baseline (T0) and after 16 (T16), 32 (T32), and 48 weeks (T48) of treatment.

### 2.3. Objectives

The aim of the study was to assess a possible correlation between the clinical outcome of dupilumab therapy and baseline BMI. Moreover, we wanted to assess potential weight gain during treatment with dupilumab.

### 2.4. Statistical Analysis

Statistical analyses were undertaken using SPSS software (IBM, Armonk, NY, USA, version 29.0). Descriptive statistics are reported as mean and standard deviation (SD) or median and 25°–75° quartile (Q1–Q3) for quantitative variables based on the distribution of the population (symmetrical vs. asymmetrical). Absolute numbers (*n*) and frequencies (%) are used for categorical variables. Dichotomous normal distributions were compared using the Student’s *t*-test, and dichotomous non-normal distributions were tested using the Mann–Whitney U test. Categorical variables were analyzed using a chi-square test or Fisher’s exact test as appropriate. Paired samples t-tests or Wilcoxon tests were used to investigate potential differences in BMI between T0 and T48, stratifying the population by age and baseline BMI. The improvement of the scores evaluated at different time points was tested using the Wilcoxon test. Pearson’s correlation index was used to assess a potential correlation between the EASI and BMI. We performed a single regression and a multivariate regression analysis to evaluate potential baseline factors influencing an EASI improvement of less than 75% at T48. To calculate the sample size, the power*G software (version 3.1) was used. Considering a *t*-Student test for two independent samples, with a significance level of 0.05 and 80% power, and assuming an effect size of 0.5 in terms of the EASI score, we determined that a minimum sample size of 72 patients was necessary. All statistical analyses were two-tailed and performed with an alpha error of 0.05. A *p*-value < 0.05 was considered significant.

## 3. Results

Our population comprised 170 patients, of which 79 (46.5%) were women and 91 (53.5%) were men. At baseline, the mean age of the population was 40.3 years (SD 16.2). At baseline, 14 (8.2%) patients received systemic corticosteroids (CS), 3 (1.8%) received methotrexate (MTX), and 29 (17.1%) received cyclosporin (CSA). The BMI at T0 ranged from 16.2 to 37.7 with a mean of 23.6. The number of individuals with a BMI < 25 was 118 (69.4%), and individuals with a BMI ≥ 25 consisted of 52 (30.6%) patients. Within the first group, the number of individuals with a BMI < 18.5 was 8 (4.6%), and individuals with a BMI between 18.5 and 24.9 consisted of 110 (64.7%) patients. Within the second group, 43 (25.3%) patients had a BMI between 25.0 and 29.9 (overweight), 7 (4.15%) had a BMI between 30.0 and 34.9 (obesity grade I), and only 2 patients (1.2%) had a BMI over 35.

The baseline epidemiological and clinical characteristics of our overall population, classified by BMI, are summarized in Table 1. There was a significant prevalence of males in the BMI ≥ 25 group compared to the BMI < 25 group (67.3% vs. 47.5%), which reflects the percentage of females (52.5% vs. 32.7%) in the BMI < 25 group. Almost half of the patients had a classical AD phenotype (46.5%), as it is the most common worldwide. The distribution of baseline age showed a slight positive skew, with most patients being young or middle-aged adults; only 22 patients started dupilumab therapy after 60 years of age. There was no statistically significant difference regarding baseline characteristics between the two BMI groups, except for the previous use of MTX (*p* = 0.034; contingency coefficient [CC] = 0.180), sex (*p* = 0.017; CC = 0.180), onset age (*p* = 0.033), and baseline age (*p* = 0.003).

The number of patients with a BMI < 25 was 118 (69.4%), and those with a BMI ≥ 25 consisted of 52 (30.6%) patients. Since BMI varies among age and sex categories [26], BMI was assessed at T0 and T48, and the population was stratified by age groups and sex [Figure 1]. In the overall population, among patients with a BMI < 25, the mean (SD) BMI was 21.4 (2.0) at T0 and 21.6 (2.1) at T48, while among patients with a BMI ≥ 25, the mean (SD) BMI was 28.4 (2.7) at T0 and 28.8 (4.6) at T48. No significant differences were found in BMI between T0 and T48 among any age, sex, or baseline BMI groups.

Regarding the EASI score, in patients with a BMI < 25, the median (Q1–Q3) EASI at T0 was 26.0 (24.0–30.0), 6.0 (3.0–10.0) at T4, 3.0 (1.0–6.0) at T16, 2.0 (1.0–4.0) at T32, and 2.0 (1.0–4.0) at T48. Among patients with a BMI ≥ 25, the median (Q1–Q3) EASI at T0 was 26.0 (24.0–30.8), 7.0 (2.0–12.0) at T4, 4.0 (2.0–7.8) at T16, 3.0 (1.0–5.0) at T32, and 2.0 (1.0–5.0) at T48. In patients with a BMI < 25, the median (Q1–Q3) EASI percentage improvement was 79.2 (62.0–88.1) at T4, 88.2 (76.7–96.0) at T16, 92.0 (87.5–96.7) at T32, and 94.3 (87.5–97.5) at T48. Among patients with a BMI ≥ 25, the median (Q1–Q3) EASI percentage improvement was 75.5 (65.2–85.4) at T4, 86.6 (77.0–93.2) at T16, 89.3 (82.4–95.8) at T32, and 92.4 (81.4–96.9) at T48 [Figure 2]. The EASI score was significantly improved as early as the first 4-week follow-up [Appendix A]. No significant difference was found at baseline between the two BMI groups. Although the difference in the absolute EASI score and its percentage improvement between the two BMI groups is not statistically significant at any time point [Appendix A], it is inferred that the improvement in the EASI is slower in patients with a BMI ≥ 25. However, this difference is not clinically significant. No statistically significant correlation was found between BMI at T0 and the EASI at T0, or between BMI at T48 and the EASI at T48.

The percentage of patients achieving at least 90% improvement in the EASI (EASI-90) at T48 is 42.4% in the BMI < 25 group vs. 36.5% in the BMI ≥ 25 group. The percentage of patients reaching EASI-100 is similar in both groups (22.9% vs. 23.1%). The percentage of patients with a BMI ≥ 25 not reaching EASI-75 is higher (13.5% vs. 5.9%) compared to patients with an EASI < 25, but the difference is not statistically significant [Figure 3]. Despite this, according to single and multivariate regression analyses, no factor among phenotype, BMI, sex, atopic family history, onset age, the number of atopic comorbidities, or the intake of systemic drugs at baseline appeared to increase the risk of not reaching EASI-75 at T48 [Appendix A].

The trends of pruritus/sleep NRS, ADCT, POEM, and DLQI by BMI group at different time points are reported in Table 2 and in greater detail in the Appendix A. Ten patients were excluded from the ADCT analysis due to missing data at T48. All the scores evaluated were significantly improved as early as the first 4-week follow-up [Appendix A]. No statistically significant differences were found between the two BMI groups at T0 for pruritus NRS, sleep NRS, and POEM. A statistically significant difference was found at baseline for ADCT (*p* = 0.043) since the median ADCT score in the normal BMI group was slightly lower (21.5 vs. 20.0). A statistically significant difference in DLQI percentage improvement was found at T4 (*p* = 0.022) and T16 (*p* = 0.002), indicating that normal-weight patients had better median DLQI improvement at T4 (73.7 vs. 59.0), whereas overweight and obese patients had better improvement at T16 (89.2 vs. 73.6). Additionally, a statistically significant difference was found for POEM at T16 (*p* = 0.013), with overweight and obese patients showing greater score improvement [Appendix A].

## 4. Discussion

We investigated a potential difference regarding the baseline characteristics among the two BMI groups. A statistically significant difference was found for the previous use of MTX, sex, onset age, and baseline age. The difference concerning sex was due to the higher number of males in the BMI ≥ 25 group. However, there are no data in the literature regarding a different response to dupilumab between males and females. Moreover, the contingency coefficient was very close to 0 (0.180), indicating a weak association. The statistically significant difference in previous MTX intake was due to 21.2% of patients in the BMI ≥ 25 group having been treated with MTX, compared to 9.3% in the BMI < 25 group. However, the previous use does not indicate that the patients were on MTX when starting dupilumab, so there is no interference with the clinical response to dupilumab. The difference in onset age is statistically significant but not clinically significant, with medians of 1.0 vs. 3.0 years for the BMI < 25 and BMI ≥ 25 groups, respectively. When considering onset age categorically (infants, children, adolescents, and adults), the difference is not statistically significant. The statistically significant difference in baseline age reflects a gap of over 10 years between the means of the two BMI groups, due to selection bias, and there is no evidence that overweight or obese patients access biologic therapy later. No different response to dupilumab in different age groups has been demonstrated [27].

Regarding our study, no differences were found in BMI between T0 and T48 among any age, sex, or baseline BMI group, meaning that dupilumab treatment does not seem to impact weight. To our knowledge, only one report has described a possible association between weight gain and treatment with dupilumab. In a Swedish study involving 12 AD patients treated with dupilumab, all patients experienced significant weight gain (mean: 6.1 kg) after 48 weeks of follow-up [16]. However, there was no significant correlation between weight gain and treatment response, reported appetite, or sleep disturbances caused by itching. Nonetheless, a study conducted on a population of only 12 subjects has limited significance. Additionally, the dupilumab randomized phase 3 studies (SOLO 1, SOLO 2, AD ADOL, and CHRONOS) did not evaluate potential weight gain during treatment [28], and no other studies have mentioned it [29]. In contrast, our study was conducted on a statistically significant and better representative sample of the AD population.

Jung MJ et al. demonstrated how weight reduction impacts the improvement of AD symptoms. Forty subjects with AD were divided into a weight maintenance group and a weight reduction group. In the weight reduction group, there was a significant improvement in the Eczema Area and Severity Index (EASI) score (BMI and the EASI showed a positive correlation), while no significant improvement was observed in the weight maintenance group [20]. In the field of psoriasis, it has already been known for several years that weight and BMI have an impact on the clinical response to biologics, namely that optimal responses to fixed-dose biological agents are less common in patients with increasing weight [30]. BMI influences the initial clinical response to systemic treatment for psoriasis [31], and reducing body weight in obese patients receiving biologics may enhance the drug’s efficacy [32]. With regard to our study, patients with a BMI < 25 have a higher EASI percentage improvement [Figure 2], which ranges from 1.9 to 3.7 points of difference. This suggests a trend where overweight and obese patients may take longer to fully respond to dupilumab treatment. However, this numerical difference does not appear to be clinically significant, meaning it may not impact treatment decisions in practice. Our results are in line with those of Patruno et al. [33], who showed that BMI correlates with a lower effectiveness of dupilumab during the first weeks of therapy. Even if not statistically significant, the percentage of overweight and obese patients who do not reach EASI-75 is higher compared to normal-weight patients (13.5% vs. 5.9%). Despite this, the multivariate analysis found that no baseline factor, including BMI, appeared to increase the risk of not reaching EASI-75 at T48. This underscores the complexity of factors influencing treatment response in AD beyond BMI alone, including genetic predispositions, disease severity, and other clinical variables. Overall, while BMI may influence the rate of improvement in AD symptoms with dupilumab therapy, its impact appears nuanced and not a decisive factor in predicting treatment outcomes based on our findings, especially when considering other variables in the analysis.

When considering PROs, patients with a BMI < 25 showed a better DLQI improvement at T4 (even though the result is the opposite at T16) and a better POEM improvement at T16. The percentage improvement is not statistically significantly different at T48 between the two BMI groups. These findings suggest that BMI may influence certain aspects of treatment efficacy in the first weeks of treatment.

### Limitations

The limitations mainly relate to the retrospective nature of the study. In addition, patients’ weight was reported by patients using personal scales at home. Subanalyses could not be conducted within the group of patients with BMI ≥ 25 because the obese category was underrepresented numerically (*n* = 9). Our study showed some difference in the percentage improvement of the EASI that should be confirmed in a larger case series, where the subpopulations of underweight, overweight, and obese patients are better represented. Based on the sample size analysis, our sample is adequate. However future research is warranted to validate and strengthen the robustness and reliability of the findings.

## 5. Conclusions

No differences were found in BMI between T0 and T48 among any age, sex, or baseline BMI group, meaning that dupilumab treatment does not seem to impact weight. Even if not significant, patients with a BMI < 25 have a higher EASI percentage improvement, suggesting a trend where overweight and obese patients may take longer to fully respond to dupilumab treatment. However, this numerical difference does not appear to be clinically significant, meaning it may not impact treatment decisions in practice.

## Figures and Tables

**Figure 1 jcm-13-04559-f001:**
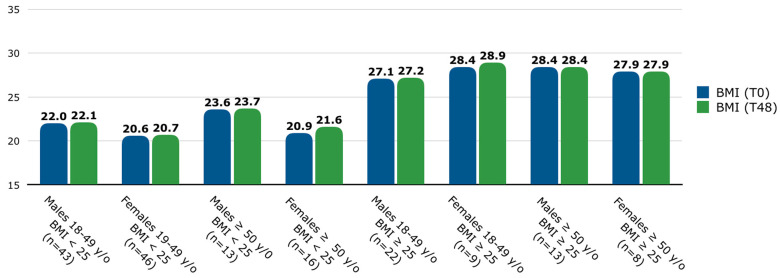
Mean BMI (at T0 and T48) stratified by age, sex, and baseline BMI groups. Patients were classified into two age groups: between 18 and 49 years old and ≥50 years old. Abbreviations: y/o, years old; BMI, body mass index; T, time point in weeks.

**Figure 2 jcm-13-04559-f002:**
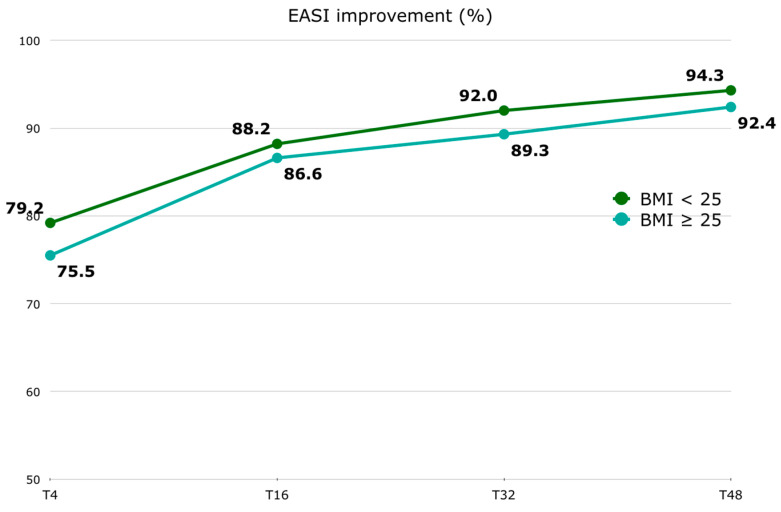
EASI percentage improvement (represented by median) trend stratified by BMI groups (BMI < 25 and BMI ≥ 25). Abbreviations: EASI, Eczema Area and Severity Score Index; BMI, body mass index; T, time point in weeks.

**Figure 3 jcm-13-04559-f003:**
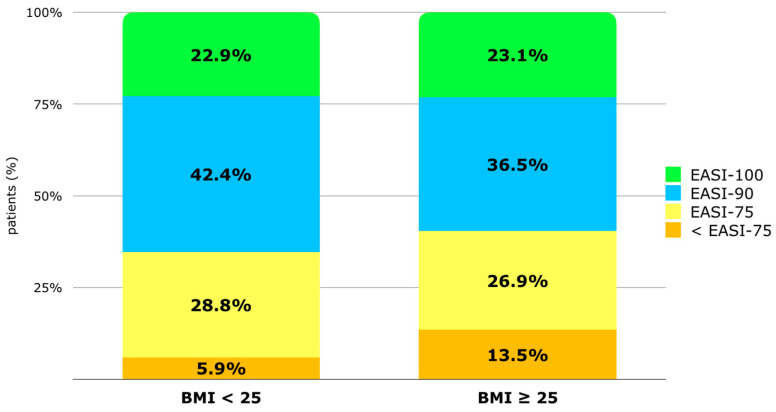
Distribution of population (%) based on EASI improvement after 48 weeks of treatment: improvement inferior to 75% from baseline (<EASI-75), improvement of at least 75% (EASI-75), EASI-90, and EASI-100. Patients are classified by BMI group (BMI < 25 and BMI ≥ 25). Abbreviations: BMI, body mass index; EASI, Eczema Area and Severity Index.

**Table 1 jcm-13-04559-t001:** Baseline clinical and epidemiological characteristics of our overall population (first column) divided into two BMI groups (second and third columns). The last columns indicate if there is a significant or non-significant statistical difference between the two BMI groups. The different apexes indicate the test that was used: ^#^ Pearson’s chi-squared; ^§^ Fisher’s exact test; CC, contingency coefficient; ^+^ independent *t*-test; and * Mann–Whitney U test. Abbreviations: BMI, body mass index; SD, standard deviation; Q1, 25° quartile; Q3, 75° quartile; AD, atopic dermatitis.

Baseline Characteristics	Overall Population	BMI < 25 (*n* = 118)	BMI ≥ 25 (*n* = 52)	*p*-Value
Baseline age, mean (SD)	40.3 (16.2)	37.8 (15.8)	45.9 (16.0)	0.003 ^+^
Sex, *n* (%) - Female - Male	79 (46.5) 91 (53.5)	62 (52.5) 56 (47.5)	17 (32.7) 35 (67.3)	0.017 ^#^ (CC = 0.180)
AD phenotype, *n* (%) - Classical - Nummular eczema - Erythrodermic - Generalized inflammatory - Generalized lichenoid - Head and neck - Hands - Prurigo nodularis	79 (46.5) 5 (2.9) 9 (5.3) 31 (18.2) 27 (15.9) 1 (0.6) 3 (1.8) 15 (8.8)	55 (46.6) 2 (1.7) 6 (5.1) 25 (21.2) 20 (16.9) 2 (1.7) 2 (1.7) 8 (6.8)	24 (46.2) 3 (5.8) 3 (5.8) 6 (11.5) 7 (13.5) 1 (1.9) 1 (1.9) 7 (13.5)	0.264 ^§^
Onset age, median (Q1–Q3)	1.0 (0.0–15.0)	1.0 (1.0–5.5)	3.0 (0.0–27.5)	0.033 *
Onset age (categories), *n* (%) - Infants (0 years old) - Children (1–12 years old) - Adolescents (13–17 years old) - Adults (≥18 years old)	71 (41.8) 52 (30.6) 8 (4.7) 39 (22.9)	54 (45.8) 38 (32.2) 4 (3.4) 22 (18.8)	17 (32.7) 14 (26.9) 4 (7.7) 17 (32.7)	0.091 ^§^
Atopic comorbidities, *n* (%) - Rhinitis - Conjunctivitis - Asthma - Food allergies - Nasal polyposis - Eosinophilic esophagitis	123 (72.4) 94 (55.3) 83 (48.8) 28 (16.5) 3 (1.8) 0 (0.0)	89 (75.4) 70 (59.3) 59 (50.0) 10 (16.1) 2 (1.7) /	34 (65.4) 24 (46.2) 24 (46.2) 9 (17.3) 1 (1.9) /	0.177 ^#^ 0.112 ^#^ 0.644 ^#^ 0.485 ^#^ 1.00 ^§^ /
Atopic family history, *n* (%)	80 (47.1)	57 (48.3)	23 (44.2)	0.624 ^#^
Previous systemic treatments, *n* (%) - Cyclosporine - Methotrexate - Azathioprine - Biological therapy	147 (86.5) 22 (12.9) 8 (4.7) 0 (0.0)	105 (89.0) 11 (9.3) 5 (4.2) /	42 (80.8) 11 (21.2) 3 (5.8) /	0.149 ^#^ 0.034 ^#^ (CC = 0.180) 0.701 ^§^ /

**Table 2 jcm-13-04559-t002:** An assessment of pruritus NRS, sleep NRS, POEM, ADCT, and DLQI, compared between the two BMI groups at different time points. The percentage improvement is represented as a median (Q1–Q3). Abbreviations: BMI, body mass index; POEM, Patient-Oriented Eczema Measure; DLQI, Dermatology Life Quality Index; ADCT, Atopic Dermatitis Control Tool; NRS, Numerical Rates Scale; T, time point in weeks.

Score	BMI Group	T0	Improvement (%) T0–T4	Improvement (%) T0–T16	Improvement (%) T0–T32	Improvement (%) T0–T48
Pruritus NRS (*n* = 170)	BMI < 25	9.0 (8.0–10.0)	60.0 (39.4–77.8)	70.0 (37.5–87.5)	70.0 (52.8–88.9)	70.0 (50.0–90.0)
	BMI ≥ 25	9.0 (8.0–10.0)	53.6 (30.0–77.8)	70.7 (55.6–87.5)	75.0 (41.1–90.0)	75.0 (34.4–90.0)
Sleep NRS (*n* = 170)	BMI < 25	8.0 (6.0–10.0)	88.9 (53.6–100.00)	100.0 (64.6–100.0)	100.0 (83.9–100.0)	100.0 (100.0–100.0)
	BMI ≥ 25	7.0 (4.3–9.0)	85.7 (50.0–100.0)	100.0 (71.4–100.0)	100.0 (77.8–100.0)	100.0 (71.0–100.0)
POEM (*n* = 170)	BMI < 25	23.0 (18.0–26.0)	61.1 (30.8–79.6)	63.6 (38.8–81.6)	75.0 (55.6–86.4)	72.1 (50.0–84.9)
	BMI ≥ 25	23.5 (20.0–28.0)	60.7 (41.3–81.9)	78.8 (54.7–86.6)	71.4 (44.2–90.7)	76.3 (58.2–89.3)
ADCT (*n* = 160)	BMI < 25	20.0 (16.8–23.0)	64.9 (50.7–75.0)	73.9 (60.0–83.3)	81.3 (70.0–92.3)	81.8 (71.4–92.3)
	BMI ≥ 25	21.5 (18.0–24.0)	62.5 (47.8–72.7)	71.4 (55.6–80.0)	77.3 (63.6–91.7)	76.5 (63.2–91.3)
DLQI (*n* = 170)	BMI < 25	15.0 (11.0–20.25)	73.7 (42.7–90.2)	73.6 (45.3–88.9)	83.3 (61.7–92.8)	83.3 (62.4–93.3)
	BMI ≥ 25	14.0 (10.0–20.0)	59.0 (28.0–79.9)	89.2 (71.2–95.0)	80.0 (50.0–93.1)	(74.0–96.1)

## Data Availability

Data are not available due to privacy restrictions.

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
