# Peer review of "Does Body Mass Index Impact the Clinical Response to Dupilumab Therapy in Atopic Dermatitis? A Monocentric Study of 170 Patients"

_jcm, 2024, doi:10.3390/jcm13154559_

Round 1

Reviewer 1 Report

Comments and Suggestions for Authors

Abstract

The abstract is well-structured and the keywords are appropriate.

Introduction

Although the introduction is well described, it might be beneficial for the authors to start discussing the effects of dupilumab on adipose metabolism rather than treating the topics of obesity and the mechanism of action of dupilumab independently.

Materials and Methods

The sample size is adequate to infer results.

Data Collection

To correctly measure the effect of BMI, it is advisable to divide according to BMI into normal weight, underweight, overweight, and obesity categories, or at least segregate into normal weight, overweight, and obesity to observe significance trends. This would enrich the authors' work.

Statistical Analysis

The statistical analysis is adequate.

Results

The results are correctly presented, although the study mainly performs descriptive analysis. It would be interesting to provide data from a tabulated bivariate analysis.

Discussion

The discussion is correct, and one of the main limitations of the study is that  patients' weight was reported by patients using personal scales at home, making the results not extrapolable.

Reviewer 2 Report

Comments and Suggestions for Authors

This study aims to investigate the effect of body mass index on the clinical response to dupilumab therapy in atopic dermatitis, the authors found overweight and obese  does not impact clinical response to dupilumab therapy. This is a study with negative results, which cannot be ruled out as the result of the small sample size of the study. In addition, the authors did not provide statistical evidence for the calculation of sample size.

Author Response

Reviewer 2

This study aims to investigate the effect of body mass index on the clinical response to dupilumab therapy in atopic dermatitis, the authors found overweight and obese does not impact clinical response to dupilumab therapy. This is a study with negative results, which cannot be ruled out as the result of the small sample size of the study. In addition, the authors did not provide statistical evidence for the calculation of sample size.

Authors comment: Dear Reviewer, thanks for your comment. You raise a valid point regarding the potential impact of the small sample size on the study's results. In our retrospective study, to calculate the sample size, the power*G software was used. Considering a t-Student test for two independent samples, with a significance level of 0.05 and 80% power, and assuming an effect size of 0.5 in term of EASI score, we determined that a minimum sample size of 72 patients was necessary for the statistical significance. Therefore, although the number of patients in this work was sufficient to determine a statistically significant difference, we are aware that future research is warranted to validate and strengthen the robustness and reliability of the findings.

 We added the information about the sample size calculation in Materials and Methods (lines 120-123 of the highlighted manuscript) and as a comment in Limitations section (lines 289-295 of the highlighted manuscript).

Reviewer 3 Report

Comments and Suggestions for Authors

Very interesting and well written article 

The data are fairly new although there are already articles like this in the literature but the sampling of this study makes it in my opinion useful for the literature 

The structure should be broken down as follows. 

Discussion and then conclusion otherwise the reader may misunderstand

Okay paragraph limitation of the study although it needs to be expanded in my opinion 

Introduction is poor in modern citations are all articles from 2018-1020, update with more recent examples like this one 

- DOI: 10.1111/dth.15901

Improving English

There are some typos in the text check the whole text

Comments on the Quality of English Language

Minor editing of English language required

Author Response

Reviewer 3

1) Very interesting and well written article. The data are fairly new although there are already articles like this in the literature, but the sampling of this study makes it in my opinion useful for the literature.

Authors comment: Thanks for your nice comment.

2) The structure should be broken down as follows. Discussion and then conclusion otherwise the reader may misunderstand

Authors comment: Thanks for the suggestion. We added the conclusion paragraph (lines 298-303 of the highlighted manuscript)

3) Okay paragraph limitation of the study although it needs to be expanded in my opinion 

Authors comment: Thanks for the comment. We expanded the limitation paragraph as it follows: The limitations mainly relate to the retrospective nature of the study. In addition, patients' weight was reported by patients using personal scales at home. Subanalyses could not be conducted within the group of patients with BMI ≥25 because the obese category was underrepresented numerically (n=9). Our study showed some difference in the percentage improvement of the EASI that should be confirmed in a larger case series, where the subpopulations of underweight, overweight and obese patients are better represented. Based on the sample size analysis, our sample is adequate. However future research is warranted to validate and strengthen the robustness and reliability of the findings.”

4) Introduction is poor in modern citations are all articles from 2018-2020, update with more recent examples like this one DOI: 10.1111/dth.15901

Authors comment: Thanks for the comment. We added the suggested article and 3 more recent article in the Introduction section.

5) Improving English. There are some typos in the text check the whole text.

Authors comment: Thank you. We checked and revised the whole text.

Reviewer 4 Report

Comments and Suggestions for Authors

Author Response

Reviewer 4

This research is original by studying the relationship between the response to Dupilumab treatment and BMI. Till now, several studies have investigated the association between AD and BMI, both in adults and in children.

Indeed, weight reduction improves the symptoms of cutaneous inflammatory disorders, such as psoriasis and AD. As a result, it was somewhat expected that patients with a lower BMI would have a better response to the treatment. Therefore, I appreciate the authors' effort in supporting this hypothesis.

However, it is unclear the duration of Dupilumab treatment for those 170 patients included in the study. The authors mentioned that the duration was at least 48 weeks. I couldn’t find any other information

Authors comment: Dear Reviewer, thanks for your valuable comment. We are sorry if some confusion was made. We confirm that we included patients that were treated with dupilumab for at least 48 weeks, meaning that we excluded patients that did not reach this follow-up point or discontinued the treatment before. The patients included in this study obviously continued the treatment after the first 48 weeks, but we did not report the results after 48 weeks because we wanted to focus on the possible influence on BMI in the first year of treatment. We tried to make it clearer in the methods (lines 81-82 of the highlighted manuscript).

Round 2

Reviewer 1 Report

Comments and Suggestions for Authors

All the comments have been properly adressed